# RCDN: Towards Robust Camera-Insensitivity Collaborative Perception via Dynamic Feature-based 3D Neural Modeling

**Tianhang Wang**
Tongji University
tianya_wang@tongji.edu.cn

**Fan Lu**
Tongji University
lufan@tongji.edu.cn

**Zehan Zheng**
Tongji University
zhengzehan@tongji.edu.cn

**Guang Chen**[*]
Tongji University
guangchen@tongji.edu.cn

**Changjun Jiang**
Tongji University
cjjiang@tongji.edu.cn

## Abstract

Collaborative perception is dedicated to tackling the constraints of single-agent perception, such as occlusions, based on the multiple agents' multi-view sensor inputs. However, most existing works assume an ideal condition that all agents' multi-view cameras are continuously available. In reality, cameras may be highly noisy, obscured or even failed during the collaboration. In this work, we introduce a new robust camera-insensitivity problem: *how to overcome the issues caused by the failed camera perspectives, while stabilizing high collaborative performance with low calibration cost?* To address above problems, we propose **RCDN**, a **R**obust **C**amera-insensitivity collaborative perception with a novel **D**ynamic feature-based 3D **N**eural modeling mechanism. The key intuition of RCDN is to construct collaborative neural rendering field representations to recover failed perceptual messages sent by multiple agents. To better model collaborative neural rendering field, RCDN first establishes a geometry BEV feature based time-invariant static field with other agents via fast hash grid modeling. Based on the static background field, the proposed time-varying dynamic field can model corresponding motion vectors for foregrounds with appropriate positions. To validate RCDN, we create OPV2V-N, a new large-scale dataset with manual labelling under different camera failed scenarios. Extensive experiments conducted on OPV2V-N show that RCDN can be ported to other baselines and improve their robustness in extreme camera-insensitivity settings.

## 1 Introduction

Multi-agent collaborative perception[1–5] obtains better and more holistic perception by allowing multiple agents to exchange complementary perceptual information. This field has the potential to effectively address various persistent challenges in single-perception, such as occlusion[6, 7]. The associated techniques and systems also process significant promise in various domains, such as the utilization of multiple unmanned aerial aircraft for search and rescue operations[8–10], the automation and mapping of multiple robots[11–13]. As an emerging field, the research of collaborative perception faces several issues that need to be addressed. These challenges include the need for high-quality datasets[14–17], the formulation of models that are agnostic to specific tasks and models[18, 19], and the ability to handle pose error and adversarial attacks[20, 21].

---

[*]Corresponding Author. Our code is available at: https://github.com/ispc-lab/RCDN.

38th Conference on Neural Information Processing Systems (NeurIPS 2024).

However, a vast majority of existing works do not seriously account for the harsh realities[22, 23] of real-world sensors in the collaboration, such as blurred, high noise, interruption and even failure. These factors directly undermine the basic collaboration premise[24, 25] of reconstructing the holistic view based on the multi-view sensors that severely impact the reliability and quality of collaborative perception process. This raises a critical inquiry: *how to overcome the issues caused by the failed cameras' perspectives while stabilizing high collaborative performance with low calibration cost?* The designation *camera insensitivity* overcomes the unpredictable essence of the specific failure camera numbers and time; see Figure 1 for an illustration. To address this issue, one viable solution is adversarial defense[26]. By robust defense strategy, adversarial defense bypasses camera insensitivity among blurred and noise. However, its performance is suboptimal[27] and has been shown to be particularly vulnerable to noise ratios[20] and failed camera numbers.

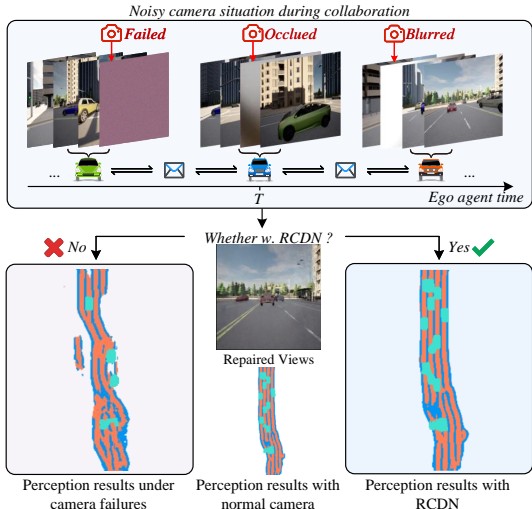

Figure 1: Illustration of noisy camera situations (blurred, occluded and even failed) during collaboration and the perception result *w.o./w.* RCDN. orange for drivable areas segmentation, blue for lanes and teal for dynamic vehicles.

To address this robust camera insensitivity collaborative perception problem, we propose **RCDN**, a **R**obust **C**amera-insensitivity collaborative perception with a **D**ynamic feature-based 3D **N**eural modeling mechanism. The core idea is to recover noisy camera perceptual information from other agents' views by modeling the collaborative neural rendering field representations. Specifically, RCDN has two collaborative field phases: a time-invariant static background field and time-varying dynamic foreground field. In the static phases, RCDN sets other baselines' backbone as the collaboration base and undertakes end-to-end training to create a robust unified geometry Bird-eye view (BEV[28, 29]) feature space for all agents. Then, the geometry BEV feature combines the hash grid modeling, an explicit and multi-resolution network, to generate static background views through $\alpha$-composed accumulation of RGB values along a ray at a fast speed. In the dynamic phase, RCDN utilizes 4D spatiotemporal position features to model the dynamic motion of 3D points, which learns an accurate motion field under optical priors and spatiotemporal regularization. The proposed RCDN has two major advantages: i) RCDN can handle camera insensitivity collaboration under unknown noisy timestamps and numbers; ii) RCDN does not put any extra communication burden into inference stage and costs little computation burden.

In our efforts to validate the effectiveness of RCDN, we identified a gap: the lack of a comprehensive collaborative perception dataset that accounts for different camera noise scenarios. To address this, we create the OPV2V-N, an expansive new dataset derived from OPV2V, featuring meticulously labeled timestamps and camera IDs. This advancement aims to support and enhance research in camera-insensitive collaborative perception. Extensive experiments on OPV2V-N show RCDN's remarkable performance when other baselines equipped with RCDN under extreme camera-insensitivity setting, improving *w.o.* RCDN baseline methods by about $157.91\%$.

## 2   Related Works

**Robust Single Perception.**   Single-agent perceptions[30, 31, 27, 32–34] have tackled the robust camera setting with other sensor modals. [27] reveals that camera-based methods [34] can be easily effected by camera working conditions. Some works[32, 31] introduce LiDAR into perception system and design a soft-association mechanism between the LiDAR and the inferior camera-side, to relieve the negative impacts caused by cameras. MVX-Net[33] improves the combination pipeline of LiDAR and cameras by leveraging the VoxelNet[35] architecture. CRN[30] introduces the low-cost Radar to replace the LiDAR, which can provide precise long-range measurement and operates reliably in all environments. However, as for the camera-only situation, few work seeks to solve this because recovering just from the single-view is highly ill-posed (with infinitely many solutions that match the

input image). With the recent rapid development of V2X[36], we now can introduce the multi-agent and multi-view based collaborative perception setting to explore this extreme situation.

**Collaborative Perception.** Perception tasks for single agents can be adversely affected by factors such as limited sensor fields of view and physical ambient occlusions. To address the aforementioned challenges, collaborative perception[37–39] can attain more comprehensive perceptual output by exchanging perception data. Early techniques involved the transmission of either unprocessed sensory input (referred to as early fusion) or the results of perception (referred to as late fusion). Nevertheless, recent research has been examining the transfer of intermediate features to achieve a balance between performance and bandwidth. Some works[40–43] devote selecting the most informative messages to communicate. DiscoNet[44] utilizes knowledge distillation to achieve a better trade-off between performance and bandwidth. V2X-ViT[45] presents a unified V2X framework based on Transformer that takes into account the heterogeneity of V2X system. Meanwhile, some learnable or mathematical based methods[46–49] have also been proposed to correct the pose errors and latency. Moreover, some works[50, 51] reveal that the holistic character of collaborative perception can improve the effect of driving planning and control tasks. However, most existing papers do not take the harsh realities of real-world sensors into account, such as blurred, high noise, occlusion and even failure, which directly undermine the basic collaboration premise of multi-view based modeling, negatively impacting performance. This work formulates camera-insensitivity collaborative perception, which considers real-world camera sensor conditions.

**Neural Rendering.** Neural radiance fields[52] aim to utilize implicit neural representations to encode densities and colors of the scene. This approach takes advantage of volumetric rendering to synthesize views, and it can be effectively optimized from 2D multi-view images. Hence, numerous works have enhanced NeRF in terms of rendering quality[53–55], efficiency[56–59], *etc*. For example, Mip-NeRF[60] utilizes cone tracing instead of ray tracing in standard NeRF volume rendering by introducing integrated positional encoding, which greatly improves the render quality. To improve the efficiency of training and inference processes, Instant-NGP[61] proposes a learned parametric multi-resolution hash for efficient encoding, which also leads to high compactness. Some works have also extended NeRF to large-scale urban autonomous scenes[62–64]. In this work, we first introduce neural rendering to collaborative perception. The proposed collaborative neural rendering field representations will address the problem of recovering highly noisy perceptual messages.

## 3 Problem Formulation

Consider $N$ agents in a scene, where each agent can send and receive collaboration messages from other agents. For the $n$-th agent, let $\mathcal{X}_n^{t_i} = \{\mathcal{I}_c^{t_i}\}_{c=1}^{c_n}$ and $\mathcal{Y}_n^{t_i}$ be the raw observation and the perception ground-truth at time current $t_i$, respectively, where $\mathcal{I}_c^{t_i}$ is the $c$-th camera images recorded at $i$-th timestamp, and $\mathcal{P}_{m \to n}^{t_i}$ is the collaboration message sent from the agent $m$ at time $t_i$. The key of the camera insensitivity is that the specific noisy camera number and corresponding timestamp are unpredictable. Therefore, each agent has to encounter invalid view information, which contains both local observation and collaboration messages sent from other agents. Then, the task of camera insensitivity collaborative perception is formulated as:

$$\max_{\theta_1, \theta_2, \mathcal{P}} \sum_{n=1}^{N} g\left(\widehat{\mathbf{Y}}_n^{t_i}, \mathbf{Y}_n^{t_i}\right) \tag{1}$$
$$\text{subject to } \widehat{\mathbf{Y}}_n^{t_i} = \boldsymbol{c}_{\theta_2}(\boldsymbol{\pi}_{\theta_1}(\psi(\mathcal{X}_n^{t_i}, \{\mathcal{P}_{m \to n}^{t_i}\}_{m=1}^{N-1}))),$$

where $g(\cdot, \cdot)$ is the perception evaluation metrics, $\widehat{\mathbf{Y}}_n^{t_i}$ is the perception result of the $n$-th agent at time $t_i$, $\psi(\cdot, \cdot)$ is the camera noise function to simulate the harsh realities of the real-world situation, $\boldsymbol{\pi}_{\theta_1}(\cdot)$ is the proposed collaborative neural rendering field network RCDN with trainable parameters $\theta_1$, and $\boldsymbol{c}_{\theta_2}$ is the existing collaborative perception network with trainable parameters $\theta_2$. Note that the proposed RCDN is to recover the noisy camera views caused by the $\psi$ function, making collaborative perception system more robust to the unpredictable situation of noisy camera data.

Given such high noisy camera view, the performances of collaborative perception system would be significantly degraded since the mainstream collaborative perception utilizes the multi-view camera-based BEV features for communication and downstream tasks, and using such damaged features

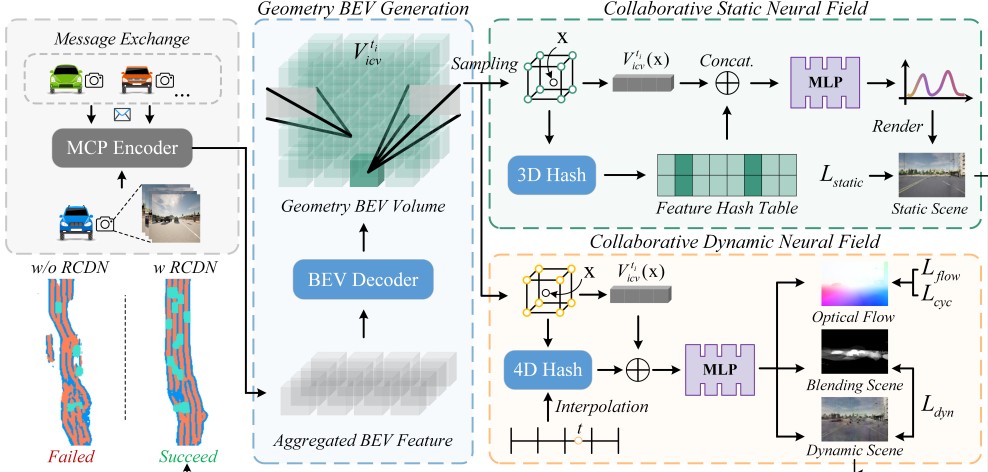

Figure 2: System overview. The geometry BEV generation module provides feature sampling for later processes. The collaborative static and dynamic fields are performed in parallel to model the background and foreground, respectively. Note that MCP is short for the multi-agents collaborative perception process.

would contain erroneous information during the perception process. In the next section, we will introduce RCDN to address this issue.

## 4 RCDN

This section proposes a robust camera-insensitivity collaborative perception system, RCDN. Figure 2 overviews the framework of the RCDN module in Sec.4.1. The details of three key modules of RCDN can be found in Sec.4.2-4.4.

### 4.1 Overall Architecture

The problem of noisy camera view results in the sub-optimization of the holistic multi-view based BEV features generation in the collaboration messages. That is, the collaboration messages from both self and other agents would be noisy or damaged for the fusion process. The proposed RCDN addresses this issue with two key notions: i) we construct novel collaborative neural rendering field representations, enabling collaborative perception to recover from the noisy camera view; and ii) we establish time-invariant and time-varying fields for background and foreground, respectively, making the collaborative neural rendering field more accurate.

Mathematically, let the $n$-th agent be the ego agent and $\mathcal{X}_n^{t_i}$ be its raw observation at the $t_i$ timestamp of agent $n$. The proposed camera-insensitivity collaborative perception system RCDN is formulated as follows:

$$\mathbf{F}_n^{t_i} = f_{\text{enc}}(\psi(\mathcal{X}_n^{t_i}, \{\mathcal{X}_j^{t_i}\}_{j=1}^{N-1})), \tag{2a}$$

$$\mathbf{V}_{icv}^{t_i} = f_{\text{geo\_bev}}(\mathbf{F}_n^{t_i}), \tag{2b}$$

$$(\sigma^s, \mathbf{c}^s) = f_{\text{static}}(\mathbf{r}(u_k), \mathbf{V}_{icv}^{t_i}(\mathbf{r}(u_k))), \tag{2c}$$

$$(\mathbf{s}_{fw}, \mathbf{s}_{bw}, \sigma_{t_i}^d, \mathbf{c}_{t_i}^d, b) = f_{\text{dynamic}}(\mathbf{r}(u_k), \mathbf{V}_{icv}^{t_i}(\mathbf{r}(u_k)), t_i), \tag{2d}$$

$$\widetilde{\mathcal{X}}_n^{t_i}, \{\widetilde{\mathcal{X}}_j^{t_i}\}_{j=1}^{N-1} = f_{\text{render}}(\sigma^s, \mathbf{c}^s, \sigma_{t_i}^d, \mathbf{c}_{t_i}^d, b), \tag{2e}$$

$$\widehat{\mathbf{Y}}_n^{t_i} = f_{\text{mcp}}(\widetilde{\mathcal{X}}_n^{t_i}, \{\widetilde{\mathcal{X}}_j^{t_i}\}_{j=1}^{N-1}), \tag{2f}$$

where $\mathbf{F}_n^{t_i} \in \mathbb{R}^{C \times H \times W}$ is the BEV feature maps of the $n$-th agent at timestamp $t_i$ with $H, W$ the size of BEV map and $C$ the number of channels; $\mathbf{V}_{icv}^{t_i} \in \mathbb{R}^{C \times Z \times H \times W}$ is the implicit collaborative geometry volume feature of the scenarios; which is lifted from BEV plane with the $Z$ height; $\mathbf{r}(u(k))$ is the ray from the failed camera center $\mathbf{o} \in \mathbb{R}^2$ through a given pixel on the image plane as $\mathbf{r}(u(k)) = \mathbf{o} + u(k)\mathbf{d}$, where $\mathbf{d} \in \mathbb{R}^3$ is the normalized viewing direction; $f_{\text{static}}$ is a explicit hash grid based representation to model the collaborative static scenarios volume density $\sigma^s \in \mathbb{R}^1$ and corresponding color $\mathbf{c}^s \in \mathbb{R}^3$; $f_{\text{dynamic}}$ is the dynamic collaborative neural network

takes the interpolated 4D-tuple $(\mathbf{r}(u(k)), t_i)$ and sampled $\mathbf{V}_{icv}^{t_i}$ feature as input and predict 3D collaborative scene flow vectors $\mathbf{s}_{fw}, \mathbf{s}_{bw} \in \mathbb{R}^3$, dynamic volume density $\sigma_{t_i}^d$, color $\mathbf{c}_{t_i}^d$ and blending weight $\mathbf{b} \in \mathbb{R}^2$; and $\widetilde{\mathcal{X}}_n^{t_i}, \{\widetilde{\mathcal{X}}_j^{t_i}\}_{j=1}^{N-1}$ is the recovered noisy camera images at timestamp $t_i$ after collaborative rendering; and $\widehat{\mathbf{Y}}_n^{t_i}$ is the final output of the system. In summary, Step 2a extracts BEV perceptual features from observation data. Step 2b generates the collaborative geometry BEV volume feature map for each timestamp, enabling feature sampling in Step 2c and 2d. Step 2d models the static background field of collaboration scenarios. Step 2d models the dynamic foreground field of collaboration objects. Step 2e gets the global volume density and color information by combining both static and dynamic field models to recover the failed camera perspective images. Finally, Step 2f outputs the final perceptual results with repaired images.

Note that i) Step 2a is done locally, Step 2b-2f are performed after receiving the messages from others. The proposed RCDN does not require any extra transmission during the inference process, which is bandwidth friendly; and ii) Step 2c and 2d are performed in parallel to save inference time; and iii) Same as [44, 49], RCDN adopts the feature representations in bird's eye view (BEV), where the feature maps of all agents are projected to the same global coordinate system. We now elaborate on the details of Steps 2b-2e in the following subsections.

## 4.2 Collaborative Geometry BEV Volume Feature

Given the BEV feature map of each agent, Step 2b aims to construct a unified collaborative geometry BEV volume feature for each timestamp of the scenario. The intuition is that [65] points out that combing with generic feature representations can avoid the per-scene "network memorization" phenomenon[52], which will improve the efficiency of the optimization process. Therefore, using the geometry BEV feature can enable the subsequent Step 2c, 2d to learn more generic networks for both static and dynamic collaborative neural fields, respectively.

To implement, we use a geometry-aware decoder $D_{geo}$ to transform the BEV feature $\mathbf{F}_n^{t_i}$ into the intermediate feature $\mathbf{F}_n^{'t_i} \in \mathbb{R}^{C \times 1 \times X \times Y}$ and $\mathbf{F}_{height,n}^{t_i} \in \mathbb{R}^{1 \times Z \times X \times Y}$, and this feature is lifted from BEV plane to an implicit collaborative volume feature $\mathbf{V}_{icv}^{t_i} \in \mathbb{R}^{C \times Z \times X \times Y}$:

$$\mathbf{V}_{icv}^{t_i} = \text{sigmoid}(\mathbf{F}_{height,n}^{t_i}) \cdot \mathbf{F}_n^{'t_i}, \tag{3}$$

where $\cdot$ represents dot production along the channel. Eq. 3 lifts the items on the BEV plane into 3D collaborative volume with the estimated height position $\text{sigmoid}(\mathbf{F}_{height,n}^{t_i})$. $\text{sigmoid}(\mathbf{F}_{height,n}^{t_i})$ represents whether there is an item at the corresponding height. Ideally, the collaborative volume feature $\mathbf{V}_{icv}^{t_i}$ contains all the scene items information in the corresponding position.

## 4.3 Static Collaborative Neural Field

After getting the collaborative volume feature $\mathbf{V}_{icv}^{t_i}$, Step 2c aims to construct the background of camera views with the static collaborative neural field. Given an arbitrary 3D scenario position $\mathbf{x} \in \mathbb{R}^3$ and a 2D viewing direction $\mathbf{d} \in \mathbb{R}^2$, we aims to estimate static scenarios volume density $\sigma^s$ and emitted RGB color $\mathbf{c}^s$ using the fast hash grid-based [61] neural network:

$$(\mathbf{c}^s, \sigma^s) = \text{MLP}(\mathbf{G}_\theta^s(\text{contract}(\mathbf{x}), \mathbf{d}); f), \quad f = \mathbf{V}_{icv}^{t_i}(\mathbf{x}), \tag{4}$$

where $f = \mathbf{V}_{icv}^{t_i}(\mathbf{x})$ is the neural feature trilinearly interpolated from the collaborative geometry BEV volume $\mathbf{V}_{icv}^{t_i}$ at the location $\mathbf{x}$, $\mathbf{G}_\theta^s(\cdot, \cdot)$ is explicit multi-level hash grid representation with the generic $f$ features for fast static collaborative neural field training. Meanwhile, owing to the collaborative scenarios are unbounded, we utilize $\text{contract}(\cdot)$[53] to map 3D scenario position into a bounded ball of radius 2 with regularization, making the estimation optimization process faster and better. Hence, we can compute the color of the pixel (corresponding to the ray $\mathbf{r}(u_k)$ using numerical quadrature for approximating the collaborative volume rendering interval[66]:

$$\mathbf{C}^s(\mathbf{r}) = \sum_{k=1}^{K} T^s(u_k)\alpha^s(\sigma^s(u_k)\delta_k)\mathbf{c}^s(u_k), \tag{5a}$$

$$T^s(u_k) = \exp\left(-\sum_{k'=1}^{k-1} \sigma^s(u_k)\delta_k\right), \tag{5b}$$

where $\alpha^s(x) = 1 - \exp(-x)$ and $\delta_k = u_{k+1} - u_k$ is the distance between two quadrature points. The $K$ quadrature points $\{u_k\}_{k=1}^K$ are drawn uniformly between $u_n$ and $u_f$, which denotes the near and far of the bounded collaborative scenarios. $T^s(u_k)$ indicates the accumulated transmittance from $u_n$ to $u_k$. Here, we denote $\mathbf{r}_i$ as the rays passing through the pixel $i$. Then, the collaborative static neural loss $\mathcal{L}_{static}$ is defined to minimize the $l_2$-loss between the estimated colors $\mathbf{C}^s(\mathbf{r}_i)$ and the ground truth colors $\mathbf{C}^{gt}(\mathbf{r}_i)$ in the static regions (where $\mathbf{M}(\mathbf{r}_i) = 0$):

$$\mathcal{L}_{static} = \sum_i \|\mathbf{C}^s(\mathbf{r}_i) - \mathbf{C}^{gt}(\mathbf{r}_i) \cdot (1 - \mathbf{M}(\mathbf{r}_i))\|_2^2 \qquad (6)$$

## 4.4 Dynamic Collaborative Neural Field

While the static collaborative neural field is being modeled, Step 2d is building the dynamic collaborative neural field to construct the foreground of camera views. Our dynamic collaborative neural field takes 4D spatiotemporal position features as input to model dynamic motion of 3D scene flow $\mathbf{s}_{fw}, \mathbf{s}_{bw}$, volume density $\sigma_{t_i}^d$, color $\mathbf{c}_{t_i}^d$ and blending weight $\mathbf{b}$ (Note that blending weights learns how to blend the results from both the static and dynamic collaborative neural fields in an unsupervised manner, avoiding background's structure and appearance conflict the moving objects.):

$$(\mathbf{s}_{fw}, \mathbf{s}_{bw}, \mathbf{c}_{t_i}^d, \sigma_{t_i}^d, \mathbf{b}) = \mathrm{MLP}(\Delta(\mathbf{G}_\theta^d(\mathrm{contract}(\mathbf{x}), \mathbf{d}), t_i); f)), \quad f = \mathbf{V}_{icv}^{t_i}(\mathbf{x}), \qquad (7)$$

where $G_\theta^d$ shares the same hash grid representations, but for the dynamic collaborative neural field optimization; $\Delta(\cdot, \cdot)$ is the temporal interpolation functions, which makes the MLP can efficiently learn the features between keyframes in a scalable manner. Meanwhile, to improve the temporal consistency of the proposed field, we compute the collaborative scene flow neighbors $\mathbf{r}(u_k) + \mathbf{s}_{fw}$ and $\mathbf{r}(u_k) - \mathbf{s}_{bw}$ with the predicted collaborative scene flow $\mathbf{s}_{fw}, \mathbf{s}_{bw}$ to warp the collaborative neural field from the neighboring time instance to the current time. Note that the term $\mathbf{s}_{fw}$ stands for forward scene flow, while $\mathbf{s}_{bw}$ refers to backward scene flow. Specifically, the forward scene flow ($\mathbf{s}_{fw}$) estimates the flow from time t to t+1, whereas the backward scene flow ($\mathbf{s}_{bw}$) estimates the flow from time t to t-1. Hence, we can obtain the corresponding density and color of adjacent time by querying the same MLPs model at $\mathbf{r}(u_k) + \mathbf{s}$:

$$(\mathbf{c}_{t_i+1}^d, \sigma_{t_i+1}^d) = \mathrm{MLP}(\Delta(\mathbf{G}_\theta^d(\mathrm{contract}(\mathbf{x} + \mathbf{s}_{fw}), \mathbf{d}), t_i + 1)) \qquad (8a)$$
$$(\mathbf{c}_{t_i-1}^d, \sigma_{t_i-1}^d) = \mathrm{MLP}(\Delta(\mathbf{G}_\theta^d(\mathrm{contract}(\mathbf{x} - \mathbf{s}_{bw}), \mathbf{d}), t_i - 1)) \qquad (8b)$$

We can compute the color of a dynamic pixel of collaborative view at time $t_i$. Hence, with both the static and dynamic collaborative neural fields model, we can easily compose them into a complete model using the predicted blending weight $\mathbf{b}$ and render full color $\mathbf{C}^{full}(\mathbf{r})$ frames at noisy views and time. We utilize the following approximate of collaborative volume rendering integral:

$$\mathbf{C}_{t_i}^{full}(\mathbf{r}) = \sum_{k=1}^K T_{t_i}^{full} \left(\alpha^d(\sigma_{t_i}^d \delta_k)(1 - \mathbf{b})\mathbf{c}_{t_i}^d + \alpha^s(\sigma^s \delta_k)\mathbf{b}\mathbf{c}^s\right) \qquad (9)$$

Similar to the static collaborative rendering loss, we train the dynamic collaborative neural model by minimizing the $l_2$ reconstruction loss under time unit $\tau = \{t_i, t_i - 1, t_i + 1\}$:

$$\mathcal{L}_{dyn} = \sum_{t \in \tau} \sum_i \|(\mathbf{C}_t^{full}(\mathbf{r}_i) - \mathbf{C}^{gt}(\mathbf{r}_i))\|_2^2 \qquad (10)$$

To reduce the amount of ambiguity caused by the sparse views during collaborative perception process, we construct motion matching loss to constrain the proposed dynamic collaborative neural field. As we do not have direct 3D supervision for predicted collaborative scene flow from the motion MLP model, we utilize 2D optical flow $\boldsymbol{f}$ as indirect supervision. Specifically, we first use the estimated collaborative scene flow to obtain the corresponding 3D point. Then, we project these 3D points onto the 2D reference frame with $\varphi(\cdot)$ function. Hence, we can compute the projected collaborative scene optical flow and enforce it to match the estimated optical flow as follows:

$$\mathcal{L}_{opt} = \sum_i \left(\varphi(\boldsymbol{s}_{\{bw, fw\}}(\mathbf{r}_i)) - \boldsymbol{f}_{\{bw, fw\}}^{gt}(\mathbf{r}_i)\right) \qquad (11)$$

Meanwhile, we also regularize the consistency of the collaborative scene flow by minimizing the cycle consistency loss $\mathcal{L}_{cyc}$. See more details in the Appendix B.7.

Table 1: Map-view segmentation of different baseline methods *w.o/w* the proposed RCDN on the OPV2V-N camera-track with one random noisy camera failure in the testing phase. We report IoU for all classes.

| Model / Metric | Static Part (*Perf. Comparison*) | | | | | Dynamic Part Vehicle | |
|---|---|---|---|---|---|---|---|
| | Drivable Area | | Lane | | | | |
| | Normal | Failure *w.o/w.* RCDN | Normal | Failure *w.o/w.* RCDN | Normal | Failure *w.o/w.* RCDN | |
| F-Cooper[1] | 45.44 | 28.87/44.89(↑55.49%) | 33.17 | 15.95/32.23(↑102.07%) | 63.33 | 29.70/61.76(↑107.95%) | |
| AttFuse[16] | 45.59 | 27.99/44.38(↑58.56%) | 33.76 | 18.77/31.50(↑67.82%) | 54.14 | 24.76/52.15(↑110.62%) | |
| DiscoNet[44] | 42.30 | 24.31/38.54(↑58.54%) | 24.24 | 12.29/22.97(↑86.90%) | 46.56 | 9.25/43.03(↑365.19%) | |
| V2VNet[37] | 41.70 | 27.99/39.72(↑41.91%) | 27.14 | 10.52/25.24(↑139.92%) | 42.57 | 11.28/42.76(↑279.08%) | |
| CoBEVT[6] | 51.96 | 32.08/47.19(↑47.10%) | 34.19 | 14.45/29.55(↑104.50%) | 56.61 | 32.41/55.10(↑70.01%) | |

## 4.5 Training Details and Optimization

To train the overall system, we supervise two tasks: static and dynamic collaborative neural fields, respectively. Meanwhile, during the training process, the static collaborative field and dynamic collaborative field are trained separately. The initial learning rate is 5e-4 with the exponential learning rate decay strategy. The weight values are set to 1.0, 1.0, 0.1, and 1.0, respectively:

$$\mathcal{L}_{total} = \lambda_1 \mathcal{L}_{static} + \lambda_2 \mathcal{L}_{dyn} + \lambda_3 \mathcal{L}_{opt} + \lambda_4 \mathcal{L}_{cyc} \tag{12}$$

## 5 Experimental Results

We create the first camera-insensitivity collaborative perception dataset and conduct extensive experiments on OPV2V-N. To ensure the consistency of the input noisy camera data and verify the effectiveness of RCDN, we set the noisy camera data to be in the failed situation[27]. Meanwhile, the task of the experiments is map segmentation, including the performance of the vehicle, drivable area (Dr. area) and lane, totaling three classes. We utilize the Intersection over Union (IoU) between map prediction and ground truth map-view labels as the performance metric.

### 5.1 Datasets

**OPV2V-N**. To facilitate research on camera-insensitivity for collaborative perception, we propose a simulation dataset dubbed OPV2V-N. In OPV2V dataset, there is a lack of mask labels for distinguishing between foreground and background views, as well as optical flow labels for supervising the scene flow. For this purpose, we collect more data to bridge the gap between neural field and collaborative perception, leading to the new OPV2V-N datasets. Specifically, we utilize the OneFormer[67] detector to extract the foreground mask labels and mainstream RAFT[68] detector to compute the optical flow between image pairs. Meanwhile, we manually annotate which part of the performance degradation is triggered by camera failure in different scenarios. See more details in the Appendix A.

### 5.2 Quantitative Evaluation

**Benchmark comparison.** The baseline methods include F-Cooper[1], AttFuse[16], DiscoNet[44], V2VNet[37] and CoBEVT[6]. All methods use the same BEV feature encoder based on CVT[69]. To validate the portability of the RCDN, we compare different baseline methods *w.o/w.* RCDN under unpredictable camera failure settings. Table 1 shows that the map-view segmentation performance of different baseline methods *w.o/w.* the proposed RCDN with only one number random noisy camera failure in the testing phase on the OPV2V-N dataset. We see that i) for static part, each baseline method with one camera failure drops about 37.73%/42.54%/32.87% (*Avg/Max/Min*) and 52.93%/61.25%/44.40% for drivable area and lane, respectively. However, each baseline method *w.* RCDN under the same camera failure situation only decreases about 5.34%/9.17%/1.22% and 7.08%/13.59%/2.85%, respectively. Compared to the *w.o* RCDN baseline methods, RCDN can improve the performance of drivable area and lane for 52.32%/58.54%/47.10% and 100.37%/139.92%/67.82%, respectively; ii) compared to the static part, as we all know, the fusion stage in collaborative perception process needs more effort on the multi-view based BEV feature map to highlight the corresponding dynamic part. Hence, the

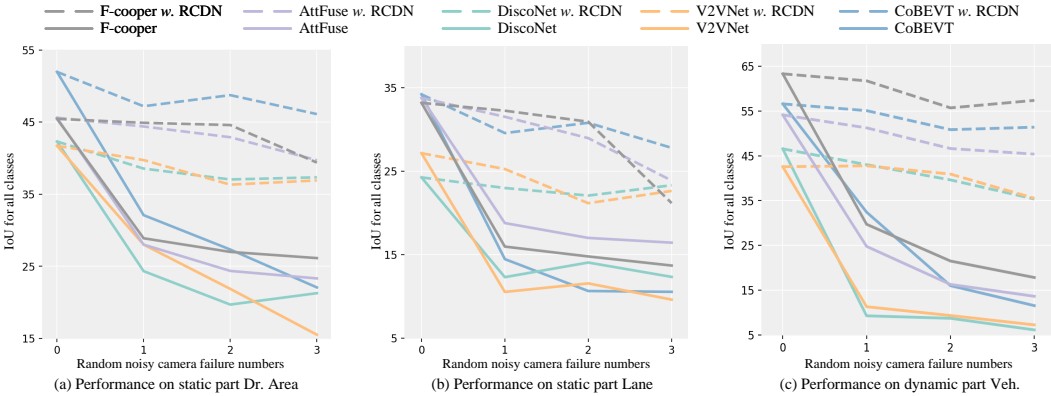

Figure 3: Comparison of the performance of other baseline methods *w.o/w* the proposed RCDN under the random noisy (failed situation) camera numbers from 0 to 3. RCDN can be ported to other baseline methods and stabilize the performance under different level camera failure situations on OPV2V-N dataset.

Table 2: Ablation Study on OPV2V-N dataset.

| Modules | | Dr. Area | Lanes | Dynamic Veh. |
| Neural Field | Time Model | | | |
| --- | --- | --- | --- | --- |
| ✗ | ✗ | 24.55 | 10.07 | 30.67 |
| ✔ | ✗ | 24.47 | 11.71 | 41.55 |
| ✔ | ✔ | 27.37 | 10.63 | 46.65 |

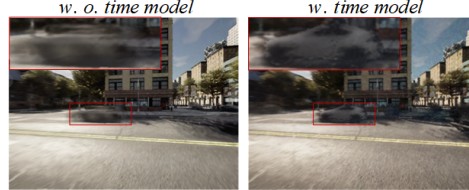

Figure 4: Effectiveness of dynamic neural field.

baseline methods' dynamic performance suffers more from camera failure than the static part, causing about a $60.75\%/42.72\%/80.14\%$ performance drop. Nevertheless, RCDN also demonstrates robustness to the dynamic foreground object modeling, with only a $3.31\%/7.58\%/0.47\%$ performance decrease for the dynamic part, improving the *w.o* RCDN baseline methods' performance by $186.57\%/365.19\%/70.01\%$. Meanwhile, as for the communication cost, similar to [44], we only utilize the $\mathbf{C}^{gt}$ labels during the training stage, meaning we leave the communication burden to the training stage and do not introduce extra information during the inference.

**Robust to extremely noisy camera data.** We conduct experiments to validate the performance under the impact of random noisy camera numbers. Figure 3 shows the map-view segmentation performance of the different baselines methods *w.o/w.* the proposed RCDN under varying levels of camera failures situation on OPV2V-N, where the $x$-axis is the expectation of the number of random failed cameras during the inference stage and $y$-axis the segmentation performance. Note that, when the $x$-axis is at 0, it represents standard collaborative perception without any camera failures. We see that i) the proposed RCDN can stabilize all the baseline methods in both static and dynamic part of map-view segmentation performance at all camera failure settings; ii) as for the static part, with the RCDN can maintain the $87.84\%/88.72\%/86.64\%$ Dr. area performance of the standard setting even under three random failed views during the collaboration process, compared with the *w.o.* RCDN only about $47.68\%/57.48\%/37.15\%$. Note that the V2VNet baseline method's performance degrades sharply as the failed camera number increases, however, with RCDN, the V2VNet can settle in a considerable performance even with the failed camera number increases; iii) as for the dynamic part, some baseline methods are crashed even with only one random camera failure situation, *e.g.* DiscoNet only maintains about $19.87\%$ performance of the standard collaborative perception setting, and almost every baseline method is unusable when there are three random camera failures, only about $20.73\%/28.11\%/13.09\%$ of the standard situation. Nevertheless, with the RCDN, we see that all baseline methods still perform well even when three random failed camera situation appear, maintaining the $84.95\%/90.81\%/75.93\%$ dynamic performance of the standard situation.

## 5.3 Qualitative Evaluation

**Visualization of segmentation.** We illustrate the map-view segmentation of other baseline methods *w.o/w.* RCDN and the corresponding repaired camera view in Figure 5. The random camera failure number is one. The orange represents the drivable area, the blue represents the lanes and the teal

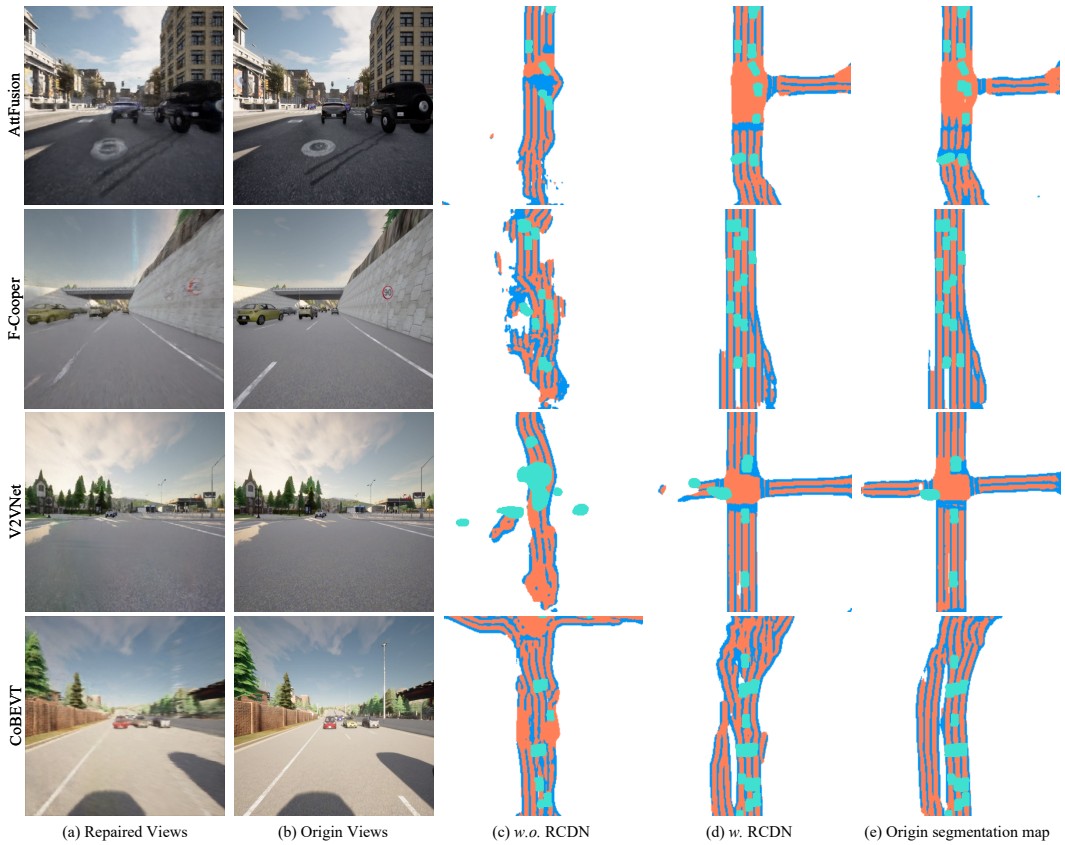

(a) Repaired Views     (b) Origin Views     (c) *w.o.* RCDN     (d) *w.* RCDN     (e) Origin segmentation map

Figure 5: Visualization of different baseline methods *w.* RCDN with one random camera failure.

represents the vehicles. We can see that i) other baseline methods show significant improvement in *w.* RCDN under noisy camera data; ii) V2VNet that collapses with noise camera data can also achieve the same level of performance as the origin data with the help of RCDN.

### 5.4 Ablation Study

**Components analysis** We conduct ablation studies on OPV2V-N with the CoBEVT baseline method. Table 2 assesses the effectiveness of the proposed two field phases. We see that i) only one neural field can recover most static part performance from the noisy camera data; ii) the proposed time model in collaborative dynamic fields can handle the motion blurry caused by the vehicles, shown in Figure 4. Meanwhile, we compare the training efficiency of the proposed RCDN with existing dynamic fields modeling methods[70], as illustrated in Figure 6. Our ap-

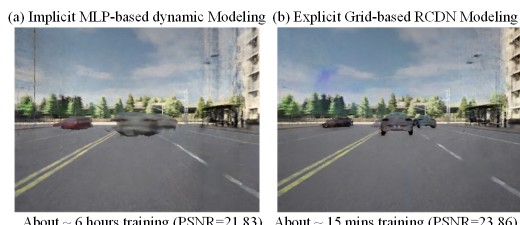

(a) Implicit MLP-based dynamic Modeling    (b) Explicit Grid-based RCDN Modeling

About ∼ 6 hours training (PSNR=21.83)    About ∼ 15 mins training (PSNR=23.86)

Figure 6: Comparison between existing dynamic field modeling and the proposed RCDN.

proach, which leverages explicit grid and geometry feature-based representations, accelerates the training process by approximately $24\times$ compared to the existing implicit MLP-based modeling, while also achieving superior PSNR quality. See more discussions in the Appendix B.2.

**Performance bottlenecks** Regarding the increasing number of agents and cameras, we validated the impact of adding more cameras using the OPV2V-N dataset (corresponding scenario types are T section and midblock respectively) with the CoBEVT baseline. From Table 3, we observe the following: i) With a single overlapping camera view, the proposed method significantly improves baseline performance, and ii) While theoretically, more cameras can provide a larger overlap range, the addition of multiple cameras (depending on their positions) may introduce redundant viewing angles, resulting in less significant performance improvements.

Table 3: Map-view segmentation performance validation about the increasing number of cameras under OPV2V-N datasets with CoBEVT baseline. Note that the failure setting is under one random noisy camera failure in the testing phases. We report IoU for all classes.

| Methods | Metrics | Scene | Failure | Overlap Cameras | | |
| --- | --- | --- | --- | --- | --- | --- |
| | | | | +1 | +2 | +3 |
| CoBEVT[6] | Dr. Area | T Section | 23.23 | 26.97 | 26.91 | 27.23 |
| | | Midblock | 23.43 | 38.87 | 38.94 | 39.51 |
| | Dyn. Vehicles | T Section | 18.83 | 40.72 | 41.38 | 42.29 |
| | | Midblock | 16.57 | 45.60 | 48.31 | 49.88 |

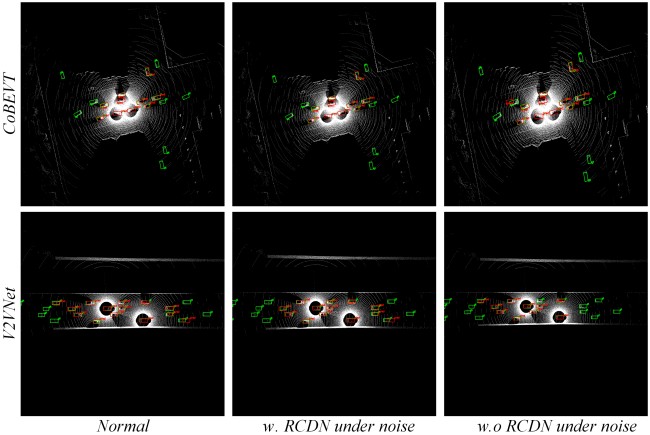

Normal     w. RCDN under noise     w.o RCDN under noise

Figure 7: Visualization of proposed RCDN for detection downstream task performance. Note that red and green boxes denote detection results and ground-truth respectively.

Table 4: Detection performance of CoBEVT and V2VNet baseline methods *w.o/w.* the proposed RCDN on OPV2V-N dataset with one random noisy camera failure in the testing phase. We report Average Precision (AP) at Intersection-over-Union (IoU) thresholds of 0.50 and 0.70.

| Methods / Metrics | Normal | |
| --- | --- | --- |
| | AP@0.50($\uparrow$) | AP@0.70($\uparrow$) |
| CoBEVT[6] | 55.56 | 43.33 |
| V2VNet[37] | 58.77 | 42.42 |
| Methods / Metrics | Failures | |
| | *w.o/w.* RCDN | |
| | AP@0.50($\uparrow$) | AP@0.70($\uparrow$) |
| CoBEVT[6] | 46.67/55.56 | 34.57/43.21 |
| V2VNet[37] | 45.45/53.85 | 36.37/38.18 |

**Different downstream tasks** Our proposed RCDN is general to different downstream tasks and is not limited to just BEV segmentation. We focus on BEV segmentation due to its crucial role in autonomous driving, with direct applications to other tasks such as layout mapping, action prediction, route planning, and collision avoidance. Additionally, we have validated RCDN for detection tasks, shown in Figure 7. We replaced the original segmentation header with a detection header in our experiments. Table 4 shows that for CoBEVT, using RCDN improves the metrics of AP@0.50 and AP@0.70 by 19.05% and 24.99%, respectively.

# 6 Conclusion and Limitation

We formulate the camera-insensitivity collaborative perception task, which considers harsh realities of real-world sensors that may cause unpredictable random camera failures during collaborative communication. We further propose RCDN, a robust camera-insensitivity collaborative perception with a novel dynamic feature-based 3D neural modeling. The core idea of RCDN is to construct collaborative neural rendering field representations to recover failed perceptual messages sent by multiple agents. Comprehensive experiments show that RCDN can be portable to other baseline methods and stabilize the performance with a considerable level under all settings and far superior robustness with random camera failures.

**Limitation and future work.** The current work focuses on addressing the camera-insensitivity problem in collaborative perception. It is evident that accurate reconstruction can compensate for the negative impact of noisy camera features on collaborative perception. In the future, we expect more works on exploring real-time collaborative neural field modeling with 3D Gaussian splatting.

# 7 Acknowledgments

This work was supported by the National Key Research and Development Program of China (No. 2021YFB2501104), in part by the National Natural Science Foundation of China (No. 62372329), in part by Shanghai Scientific Innovation Foundation (No. 23DZ1203400), in part by Tongji-Qomolo Autonomous Driving Commercial Vehicle Joint Lab Project, and in part by Xiaomi Young Talents Program.

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
