# OpenReview forum: "RCDN: Towards Robust Camera-Insensitivity Collaborative Perception via Dynamic Feature-based 3D Neural Modeling"
_NeurIPS.cc/2024/Conference — NeurIPS 2024 poster_

### Official Review · Reviewer_fQwk · 2024-07-10

**Soundness:** 2
**Presentation:** 2
**Contribution:** 2
**Rating:** 6
**Confidence:** 3

**Summary:**

In this paper, the authors proposed an essential problems: how to overcome the issues caused by the failed camera perspectives, while stabilizing high collaborative performance with low calibration cost? The authors presented a robust camera-insensitivity collaborative perception with a novel dynamic feature-based 3d neural modeling mechanism to address the issue. Moreover, to verify the effectiveness of the model, the authors also provided a new large-scale dataset, OPV2V-N for this field. The experiments result showcase the model’s robustness in proposed dataset.

**Strengths:**

Strength:
1.	The paper presents an interesting viewpoint that is to recover noisy camera perceptual information from other agents’ views by modeling the collaborative neural rendering field representation, in which the model is divided into two stages: a time-invariant static background and time-varying dynamic foreground.s
2.	The paper develops a new dataset to fill the gap of the lack of a comprehensive collaborative perception dataset that accounts for different camera noise scenarios.
3.	The paper is well-organized and interesting to read.

**Weaknesses:**

1.	From my perspective, the paper lacks the theory analysis for the proposed method. Moreover, the authors fail to introduce the motivation of each sub-module in the presented model. For example, can the authors showcase the motivation of using Nerf for the static and dynamic fields, are there any dominant advantages of nerf, compared to other 3d reconstruction methods in this method?
2.	It is necessary to give more rigorous mathematic analysis of equations in this paper. Furthermore, the authors are required to introduce the details of each networks, including the training parameters, learning rate, weight values in eq. 12.

**Questions:**

see weakness part

**Limitations:**

The current work focuses on addressing the camera-insensitivity problem in collaborative perception. It is evident that accurate reconstruction can compensate for the negative impact of noisy camera features on collaborative perception.

---

> ### Author Rebuttal · Authors · 2024-08-06
>
> Thank you for your appreciation of our work. We will take your suggestions into account when preparing the final version. Below please find responses to specific questions.
>
> >"W.1: Can the authors showcase the motivation of using Nerf for the static and dynamic fields, are there any dominant advantages of nerf, compared to other 3d reconstruction methods in this method?"
>
> **A.1** The proposed geometry BEV feature avoids the issue of per-scene “network memorization” inherent in NeRF by employing generic feature representations. The decoupling of dynamic and static neural rendering adapts to real-world collaborative autonomous scenarios. We selected NeRF for its photorealistic rendering capabilities. However, research such as Mip-NeRF 360 (CVPR 2022) and Zip-NeRF (ICCV 2023) demonstrates that other reconstruction methods, such as Structure from Motion (SfM) and Multi-View Stereo (MVS), are less effective at novel view synthesis compared to NeRF-based methods. Meanwhile, to make NeRF a better adaption to the collaborative perception setting, we optimize the collaborative neural field by decoupling it into static and dynamic parts for better performance, which improves the segmentation performance of dynamic vehicles by 12.27\% compared to the non-decoupled one.
>
> >"W.2: It is necessary to give more rigorous mathematic analysis of equations in this paper. The authors are required to introduce the details of each networks, including the training parameters, learning rate, weight values in eq. 12."
>
> **A.2** Thank you for pointing this out. The static and dynamic neural components of our model follow the architecture of Instant-NGP (SIGGRAPH 2022). The architecture for the collaborative perception component is detailed on L566-583 of the supplementary material.  The initial learning rate is 5e-4 with the exponential learning rate decay strategy. The weight values in eq. 12 is set to 1.0, 1.0, 0.1, and 1.0, respectively. Further details about the loss function are provided on L632-636 of the supplementary material. We will also include these parameters in the main text for improved clarity.
>
> >"L.1: The current work focuses on addressing the camera-insensitivity problem in collaborative perception. It is evident that accurate reconstruction can compensate for the negative impact of noisy camera features on collaborative perception."
>
> **A.1** Thanks for your deep understanding of our proposed RCDN. To the best of our knowledge, We are the first to introduce the NeRF into the collaborative perception field to handle robust settings. Our proposed RCDN has shown significant improvements in our experiments, particularly in scenarios under multi-source noises.

---

> > ### Comment · Reviewer_fQwk · 2024-08-08
> >
> > The authors have processed all my concerns. Thanks!

---

> > > ### Author Response · Authors · 2024-08-09
> > > **Thanks for your comment !**
> > >
> > > Thank you for your thorough review and constructive feedback. We sincerely appreciate your valuable reviews and are glad to know that our rebuttal and new experiments have addressed most of your concerns.

---

### Official Review · Reviewer_ymZH · 2024-07-12

**Soundness:** 2
**Presentation:** 3
**Contribution:** 2
**Rating:** 5
**Confidence:** 3

**Summary:**

The paper introduces a new problem: how to overcome the issues caused by the failed camera perspectives, while stabilizing high collaborative performance with low calibration cost? Therefore, RCDN, a Robust Camera-insensitivity collaborative perception with a novel Dynamic feature-based 3D Neural modeling mechanism is introduced. To validate the new method, the authors also provide a new dataset: OPV2V-N. RCDN serves as baseline here. Ablation Study shows for 5 models (F-cooper, Att-Fuse, Disco-Net, V2VNet, CoBEVT a significant improvement over their baselines, w/o RCDN.

**Strengths:**

The paper builds up on three pillars: single perception, collaborative perception and neural rendering. The base idea is novel to the best of my knowledge. The problem formulation is clear and well sounded, easy to follow. The System architecture is strong. The authors also focus on the differentiation between static and dynamic scenarios, especially for the neural fields both based on the BEV volume feature space. This differention is very important, not very often in detail discussed. The ablation study especially table 5.1 shows very accurate an increase of performance for different tasks static (lanes, free space) and dynamic perception. The experimentsl part introduces a new dataset, which is necessray for the investigation.

**Weaknesses:**

The overall system architecture sounds good. However, there are some open points for me, the impact of section 4.3 and 4.4, i.e. the neural fields part, seems open in terms of clarification. Example: What is difference between sf w , sbw in equation (7)?
The experimental section is a bit too short. I feel its not finished yet. However, there is limited space. The overall approach is not usable for realtime.

**Questions:**

What is difference between sf w , sbw in equation (7)?
How many message exchange tasks could be used overall (Figure 2.)  Will baseline code be publishe in combination with the dataset? When?

**Limitations:**

The most relevant limitation is the missibg real-time applicability.

---

> ### Author Rebuttal · Authors · 2024-08-06
>
> Thank you for your insightful comments. We have carefully addressed all the questions raised. Please find our responses below.
>
> >"W.1: What is difference between sfw, sbw in equation (7)?"
>
> **A.1** Apologies for any confusion regarding the terms s\_fw and s\_bw. The term s\_fw stands for forward scene flow, while s\_bw refers to backward scene flow. Specifically, the forward scene flow (s\_fw) estimates the flow from time t  to  t+1, whereas the backward scene flow (s\_bw) estimates the flow from time t  to  t-1. By leveraging both s\_fw and s\_bw, we can achieve a more consistent representation of scene movements.
>
> >"W.2: The experimental section is a bit too short. I feel its not finished yet. However, there is limited space."
>
> **A.2** Thanks for your understanding. Due to the limited space available, we aimed to present the key experimental results and analyses as concisely as possible. However, we greatly appreciate your suggestion and agree that providing additional details could enhance the clarity and comprehensiveness of our work. In response to your comment, we will make the following improvements to the experimental section in the final version: i) include a more detailed description of the experimental setup and procedures ii) add additional experimental results and figures (such as detection results) to better support our conclusions.
>
> >"Q.1: How many message exchange tasks could be used overall (Figure 2.)"
>
> **A.1** The proposed RCDN does not add extra message exchange times, and as for communication cost, similar to DiscoNet (NIPS 2021), we only utilize the RGB labels during the training stage, meaning we leave the communication burden to the training stage and do not introduce extra information during the inference.
>
> >"Q.2: Will baseline code be published in combination with the dataset? When?"
>
> **A.2** We will publish the corresponding codes and dataset as soon as we get accepted.
>
> >"L.1: The most relevant limitation is the missibg real-time applicability."
>
> **A.1** We provide the corresponding latency times in the supplementary materials: the proposed static module takes approximately 4.47 ms, the dynamic module takes about 3.94 ms, and each rendering process takes about 21 ms. The training time ranges from 20 to 30 minutes. For more details, please refer to lines L556-583. Currently, RCDN requires further code optimization to meet real-time application requirements.

---

> > ### Comment · Reviewer_ymZH · 2024-08-09
> > **Thank You**
> >
> > The authors addressed my comments very well. Thank you.
> > I would recommend to address W.1/A.1 for a final paper.
> > Overall, I stay at my decision due to the weaknesses described above.

---

> > > ### Author Response · Authors · 2024-08-09
> > > **Thanks for your comment !**
> > >
> > > Thank you for your feedback and for acknowledging our efforts to address your comments. We appreciate your recommendation regarding W.1/A.1 and will ensure that these points are carefully addressed in the final version of the paper. Your insights have been invaluable in improving our work, and we are committed to refining it further.

---

### Official Review · Reviewer_YPsN · 2024-07-12

**Soundness:** 3
**Presentation:** 4
**Contribution:** 1
**Rating:** 5
**Confidence:** 2

**Summary:**

The paper presents RCDN, a method to aggregate multi-sensor perception signals in dynamic environment.
The key idea, is to improve the aggregated multi-agent feature with the multi-view rendering loss.
At its core, RCDN gathers input streams at varying timesteps of multiple agents. The gathered images are fused into Birds Eye view (BEV) then further decoded into volume.
The volumetric features are learned into static scene and dynamic scene components with NGP based representation.
Overall procedure is supervised with rendering loss, (cyclic) optical flow consistency.


The method is evaluated on new dataset, OPV2V-N, which is an updated version of OPV2V, with additional masking and optical flow.
The results show that RCDN helps BEV segmentation with various backbones, compared to the model used without RCDN.

**Strengths:**

The main benefit of the RCDN, is that it is fairly easy to apply into different existing feature backbones, as it is the post-processing step built on top of BEV features.
Experimentally, the usage of RCDN significantly improves the segmentations which implies that the features are better aligned throughout the noisy signals.
This makes the work to be a great off-line data augmentation / preparation pipeline for generating BEV segmentation features.
The paper additionally proposes OPV2V-N dataset, which may be somewhat valuable addition to the community.

Aside from technical perspective, the paper is easy to follow and well-written.

**Weaknesses:**

The paper's main weaknesses are two folds.
1. The paper does not evaluate on tasks other than BEV segmentation.
While I believe that the pixel-aligned features from NGP would give benefits over various vision tasks, the paper only demonstrates on smaller domain of work which undermines its actual potential. It would have been more interesting to compare how it impacts in different downstreaming tasks, such as detection / tracking.

2. Technical contribution seems to lack novelty.
The paper is a mix of two known-to-work solutions; BEV feature decoding for segmentation (used with various baselines in the experiments), and NGP (or radiance field based) multi-view pixel / density alignment through rendering loss. Usage of rendering loss to improve segmentation map is well-investigated in different literatures in the NeRF community (e.g, semantic-nerf).

**Questions:**

These are few questions that I would like the authors to answer in the rebuttal.
1. How real is the synthetic OPV2V-N dataset? In other words, how can features learned in OPV2V-N dataset be translated to real-world usage? Moreover, are there any real-world quantitative results on model trained on synthetic data?

2. Have authors evaluated the method on different down streaming task other than segmentation? How does one verify that the volumetric features are geometrically correct? (how accurate is the Geometric BEV features?)

3. How is BEV segmentation evaluation differ on non-flat surfaces like hills or bridges?

**Limitations:**

No concerning limitations are found.

---

> ### Author Rebuttal · Authors · 2024-08-06
>
> Thank you for your detailed review and valuable comments. Please note our top-level comment with additional experimental and theoretical results. Below we address specific questions.
>
> >"W.1: Have authors evaluated the method on different down tasks other than segmentation?"
>
> **W.1** Our proposed RCDN is general to different downstream tasks and is not limited to just BEV segmentation. We focus on BEV segmentation due to its crucial role in autonomous driving, with direct applications to other tasks such as layout mapping, action prediction, route planning, and collision avoidance. Additionally, we have validated RCDN for detection tasks.  In our experiments, we replaced the original segmentation header with a detection header (see Figure 2 and Table 3 in the attached PDF). Table 3 shows that for CoBEVT, using RCDN improves the metrics of AP@0.50 and AP@0.70 by 19.05\% and 24.99\%, respectively.
>
> >"W.2: Technical contribution"
>
> **A.2** Robust perception is a significant challenge in single-agent systems; however, few studies have addressed this issue in the context of collaborative perception. Instead of adding additional modal sensors, such as LiDAR, as done in single-agent systems, our approach leverages the unique multi-view properties to mitigate the impact of noisy views. Notably, NeRF (Neural Radiance Fields) achieves photorealistic rendering by optimizing 2D multi-view images. To the best of our knowledge, we are the first to apply NeRF to the field of collaborative perception to handle robust settings. Meanwhile, to make NeRF a better adaption with the collaborative perception setting, we propose the geometry BEV features, which can improve the PSNR by about 9.30\%  (see more details in Appendix.B.6) and avoid NeRF network memorization. Additionally, we optimize the collaborative neural field by decoupling it into static and dynamic parts for better performance, which improves the segmentation performance of dynamic vehicles by 12.27\% compared to the non-decouple approach.
>
> >"Q.1: How real is the synthetic OPV2V-N dataset? In other words, how can features learned in OPV2V-N dataset be translated to real-world usage?"
>
> **A.1** The OPV2V-N dataset is recorded by the co-simulation with SUMO (traffic manager) under the realistic platform CARLA simulator. As for the "How can features learned in OPV2V-N dataset be translated to real-world usage?", this pertains to the broader issue of sim-to-real transfer. Whether the RCDN pre-trained on OPV2V-N can be effectively applied to other datasets or real-world scenarios depends on the domain gap between the BEV (Bird’s Eye View) feature space learned from OPV2V-N and that of other datasets or real-world environments. Recent research, such as the DUSA approach (ACM-MM 2024), addresses sim-to-real adaptation for collaborative perception. DUSA proposes a unified unsupervised BEV feature adaptation module. Since our RCDN operates as a post-processing step built upon BEV features, it is theoretically possible for RCDN to leverage DUSA to bridge the BEV feature space gap between OPV2V-N and real-world scenarios.
>
> >"Q.2: Moreover, are there any real-world quantitative results on model trained on synthetic data?"
>
> **A.2** Please refer to our top-level comment and the attached PDF for further details. In summary, we can employ sim2real BEV feature space adaptation techniques, such as DUSA, to validate models trained on synthetic data in real-world scenarios. However, the current real-world collaborative perception datasets can not meet the RCDN's camera setting demands, and we will keep track of real-world dataset development, implementing the RCDN in real-world settings if the camera setting meets the demands.
>
> >"Q.3: How does one verify that the volumetric features are geometrically correct? (how accurate is the Geometric BEV features?)"
>
> **A.3** Our proposed module only inputs RGB images without introducing additional features such as depth images. Specifically, we supervise the geometry of volumetric features by using multi-view consistency and corresponding downstream tasks. From the experimental results, the visualization results (please kindly refer to Figure 5 in the manuscript Page 9.) demonstrate the high multi-view consistency of the rendered results. The generated BEV features successfully improve the performance of perception tasks in 3D space (i.e., BEV segmentation/detection), which further indicates the geometrical correctness of our BEV feature.
>
> >"Q.4: How is BEV segmentation evaluation differ on non-flat surfaces like hills or bridges?"
>
> **A.4** Thank you for your questions. Currently, the OPV2V-N segmentation labels are available only in 2D image format. As a result, non-flat surfaces such as hills or bridges are projected onto these 2D segmentation images. Consequently, from the perspective of static part segmentation, our BEV decoder classifies static elements into two categories: road and line.

---

### Official Review · Reviewer_y98a · 2024-07-15

**Soundness:** 3
**Presentation:** 2
**Contribution:** 3
**Rating:** 5
**Confidence:** 3

**Summary:**

The paper proposed Bird Eye View (BEV) semantic segmentation pipeline from collaborative perception, robust to motion blur, sensor noise, occlusion and even failure. The proposed a pipeline that adapts neural rendering techniques to overcome the noise/malfunction in camera capture and occlusion. With the proposed method combined with prior methods, performances on OPV2V-N (the proposed BEV semantic segmentation dataset) are improved.

**Strengths:**

The paper proposed to apply neural rendering concept for ‘robust’ collaborative-perception BEV segmentation. It is natural way of thinking to overcome noise/malfunction in the caption system but the way the paper adapts neural rendering to BEV segmentation is novel. And, the performance is verified with OPV2V-N dataset.

**Weaknesses:**

Evaluation is only performed with OPV2V-N dataset which may result in overfitting. More evaluation with different dataset is required. The author may need to compare methodologies on other dataset although the existing dataset do not have noise. The author also may add random noise to the prior dataset and run experiments.

The manuscript was uneasy to read and understand. The paper should re-written. The comments below are without understanding supplemental materials fully.
- The way proposed algorithm is combined with prior method is unclear. The reviewer guessed that the MCP module can be replaced with prior methods, but it is not stated.
- Many abbreviations are not explained sufficiently and terminologies the author defined are ambiguous and may be incorrect.
- MCP is short for the multi-agents collaborative perception process but the paper did not explain MCP module in details with no reference
- BEV, no full name, no reference.
- “Camera-insensitivity” can be understood terminologies related to camera sensor sensitivity (how much the camera sensor accept photon…).
- Robust Camera-Insensitivity: Robust == Camera-sensitivity? The latter one may be redundant
- Line 6. introduce a new robust camera-insensitivity problem: cam be replaced “introduce BEV segmentation when the camera capture are unreliable (or noisy)?” Should be more concrete without ambiguous words
- Line19 “Ported to” mean?
- There are more unclear sentences.

**Questions:**

.

**Limitations:**

.

---

> ### Author Rebuttal · Authors · 2024-08-06
>
> Thank you for your detailed review and thoughtful comments. Please note our top-level comment with additional experimental and theoretical results. Below we address specific questions.
>
> >"W.1: Evaluation is only performed with OPV2V-N dataset which may result in overfitting. More evaluation with different dataset is required. "
>
> **A.1** This point was also raised by other reviewers, and we have addressed it in the general response (above). Please refer to our top-level comment and the attached PDF for detailed information. In summary, i) validated our approach on a newly collected V2XSet-N-mini dataset, demonstrating that the proposed RCDN stabilizes performance under noisy camera conditions.; ii) utilized the OPV2V-N pre-trained RCDN module for direct inference on the V2X-Sim 2.0 dataset, showing that RCDN effectively stabilizes the perception results.
>
> >"W.2: The way proposed algorithm is combined with prior method is unclear. The reviewer guessed that
> the MCP module can be replaced with prior methods, but it is not stated"
>
> **A.2** The MCP module stands for the Multi-agent Collaborative Perception module. Existing state-of-the-art (SoTA) MCP modules share a common pipeline: an encoder-fusion-decoder architecture. To ensure fairness in collaborative perception experiments, different MCP modules use the same encoder-decoder architecture but differ in the fusion process. The fusion process is responsible for the bird-eye view (BEV) feature aggregation. Therefore, the MCP module can be replaced by simply switching between different BEV feature aggregation processes.
>
> >"W.3: Many abbreviations are not explained sufficiently and terminologies the author defined are
> ambiguous and may be incorrect. MCP is short for the multi-agents collaborative perception process but the paper did not explain MCP module in details with no reference"
>
> **A.3** Thank you for pointing out the misuse of abbreviations. We will ensure to double-check and eliminate any ambiguous abbreviations. For the multi-agent collaborative perception (MCP) module, we employ mainstream collaborative methods based on intermediate BEV features for our experiments. The general setting of the MCP module is described in L570-578 supplements. To avoid any ambiguity, we will include additional technical details about the different MCP baseline methods, such as their respective pipelines and feature fusion processes, in the final supplements.
>
>
> >"W.4: BEV, no full name, no reference"\"There are more unclear sentences."
>
> **A.4** We will reintroduce the concept of bird’s-eye view (BEV) and include relevant references, such as BEVFormer (ECCV 2022). Additionally, we will address the misuse of the terms s\_fw and s\_bw, which refer to scene flow for forward and backward directions, respectively. We will also revise the paper to clarify any unclear sentences.
>
> >"W.5: “Camera-insensitivity” can be understood terminologies related to camera sensor sensitivity (how much the camera sensor accept photon…).
> "\"Robust Camera-Insensitivity: Robust == Camera-sensitivity? The latter one may be redundant
> Line 6. introduce a new robust camera-insensitivity problem: cam be replaced “introduce BEV segmentation when the camera capture are unreliable (or noisy)?” Should be more concrete without ambiguous words"
>
> **A.5** Thank you for your careful and valuable feedback. Our intent in using the term “robust” was to emphasize the concept, but we will revise the description to avoid redundancy based on your suggestions. We will replace the phrase “introduce a new robust camera-insensitivity problem” with “introduce BEV segmentation when the camera capture is unreliable (or noisy).”
>
> >"W.6: Line19 “Ported to” mean?"
>
> **A.6** Apologies for any confusion regarding the term "Ported to". The "Ported to" means that it is fairly easy to apply to different existing feature backbones, as it is the post-processing step built on top of BEV features.

---

> > ### Comment · Reviewer_y98a · 2024-08-08
> > **Thank you for the rebuttal**
> >
> > When revising the writing, it is recommended below:
> > 1. Assume that the reader is unfamiliar with the topic.
> > 2. Replace ambiguous or subjective words/sentences to more concrete and specific ones.

---

> > > ### Author Response · Authors · 2024-08-09
> > > **Thanks for your comment !**
> > >
> > > Thanks for your positive feedback and valuable insights! In our revised manuscript, we will ensure that the writing is clear and accessible, assuming that the reader may be unfamiliar with the topic. We will also replace any ambiguous or subjective language with more concrete and specific terms. Additionally, we will clearly outline all technical details to ensure a comprehensive understanding for all readers.

---

### Official Review · Reviewer_qUQv · 2024-07-15

**Soundness:** 2
**Presentation:** 3
**Contribution:** 2
**Rating:** 5
**Confidence:** 3

**Summary:**

The paper introduces RCDN, a novel method for robust camera-insensitivity collaborative perception. This method aims to overcome challenges associated with noisy, obscured, or failed camera perspectives by using dynamic feature-based 3D neural modeling. RCDN constructs collaborative neural rendering field representations to recover failed perceptual messages sent by multiple agents. The proposed system consists of two collaborative field phases: a time-invariant static background field and a time-varying dynamic field. To validate RCDN, a new dataset called OPV2V-N was created. The paper demonstrates that RCDN improves the robustness of baseline methods in extreme camera-insensitivity settings.

**Strengths:**

*Innovative Problem Addressing*: The paper tackles a significant real-world problem of camera insensitivity in multi-agent collaborative perception, which is crucial for autonomous systems.

*Novel Methodology*: The introduction of dynamic feature-based 3D neural modeling and the construction of collaborative neural rendering field representations are innovative approaches.

*Comprehensive Dataset*: The creation of the OPV2V-N dataset, which includes various camera failure scenarios, provides a robust platform for testing and validating the proposed method.

*Performance Improvement*: The extensive experiments and quantitative evaluations show significant improvements in robustness and performance over baseline methods.

*Detailed Evaluation*: The paper includes both quantitative and qualitative evaluations, along with ablation studies, which thoroughly demonstrate the effectiveness of RCDN.

**Weaknesses:**

*Complexity and Computation*: The proposed method involves complex modeling and multiple steps. The author should provide the latency.

Generalizability: The performance of RCDN is primarily validated on the OPV2V-N dataset, which may limit the generalizability of the results to other datasets or real-world scenarios.


*Failure Cases*: It would be nice if the authors provide failure cases, which is important.

**Questions:**

*Dataset Diversity*: Have you tested RCDN on any datasets other than OPV2V-N? How does it perform on real-world data?

*Real-Time Feasibility*: What are the computational requirements of RCDN, and how feasible is it for real-time applications in autonomous systems?

*Scalability*: How well does the method scale with an increasing number of agents and cameras? Are there any performance bottlenecks?

**Limitations:**

Please see weakness and questions.

---

> ### Author Rebuttal · Authors · 2024-08-06
>
> We appreciate your time in reviewing our work and your feedback on the paper's value and clarity! Please note our top-level comment with additional experimental and theoretical results. Below we address specific questions.
>
> >"Q.1: Have you tested RCDN on any datasets other than OPV2V-N? How does it perform on real-world data?"
>
> **A.1** This point was also raised by other reviewers, and we have addressed it in our general response (above). Please refer to our top-level comment and the attached PDF for detailed information. In summary, i) we validated our approach on the newly collected V2XSet-N-mini dataset, demonstrating that the proposed RCDN stabilizes performance under noisy camera conditions; ii) we also deployed the OPV2V-N pre-trained RCDN module for direct inference on the V2X-Sim 2.0 dataset, where RCDN continued to effectively stabilize perception results; iii) we will keep track of real-world dataset development, implementing the RCDN in real-world settings when the camera settings meet our requirements.
>
>
> >"W.1: The performance of RCDN is primarily validated on the OPV2V-N dataset, which may limit the generalizability of the results to other datasets or real-world scenarios."
>
> **A.1** Since the proposed RCDN operates as a post-processing step on top of BEV features, its direct applicability to other datasets or real-world scenarios depends on the domain gap between the BEV feature space learned by OPV2V-N and those of other datasets or the real world. i) for other datasets, we performed open-set validation by directly applying the OPV2V-N pre-trained RCDN module to the V2X-Sim 2.0 dataset. The corresponding results, shown in Figure 1 of the attached PDF, indicate that RCDN effectively stabilizes perception results. This is expected since both V2X-Sim 2.0 and OPV2V-N are recorded using the CARLA simulator, resulting in a minimal domain gap. ii) for real-world scenarios, the recently published DUSA (ACM-MM 2024) identified a significant domain gap between simulated and real-world environments. DUSA proposed an unsupervised BEV feature adaptation module to bridge this gap. Therefore, in theory, the proposed RCDN could leverage DUSA to mitigate the BEV feature space gap between OPV2V-N and real-world datasets.
>
> >"Q.2: What are the computational requirements of RCDN, and how feasible is it for real-time applications in autonomous systems?"
>
> **A.2** We provide the corresponding latency times in the supplementary materials: the proposed static module takes approximately 4.47 ms, the dynamic module takes about 3.94 ms, and each rendering process takes about 21 ms. The training time ranges from 20 to 30 minutes. For more details, please refer to lines L556-583. Currently, RCDN requires further code optimization to meet real-time application requirements.
>
> >"W.3: It would be nice if the authors provide failure cases"
>
> **A.3** Based on the assumption of the multi-view setting, the multi-view RCDN may fail when agents are at the edge of the communication range, resulting in minimal view overlap. This limitation motivated our effort to collect the OPV2V-N dataset for our experiments.
>
> >"Q.4: How well does the method scale with an increasing number of agents and cameras? Are there any performance bottlenecks?"
>
> **A.4** Regarding the increasing number of agents and cameras, we validated the impact of adding more cameras using the OPV2V-N dataset (corresponding scenario types are T section and midblock respectively) with the CoBEVT baseline. From Table 2 in the attached PDF, we observe the following: i) With a single overlapping camera view, the proposed method significantly improves baseline performance, and ii) While theoretically, more cameras can provide a larger overlap range, the addition of multiple cameras (depending on their positions) may introduce redundant viewing angles, resulting in less significant performance improvements.

---

### Author Rebuttal · Authors · 2024-08-06

**Please see the attached PDF for a one-page PDF with a summary of added experimental results.**

We thank all reviewers for their constructive comments on our work. We found one comment that was common amongst more than one reviewer, hence we highlight it here.

>"Have you tested RCDN on any datasets other than OPV2V-N?[qUQv]"\
>"The author may need to compare methodologies on other dataset.[y98a]"\
>"are there any real-world quantitative results on model trained on synthetic data?[YPsN]"

**A1.1** **For other datasets**, to adapt the multi-view based robust collaborative perception setting, we spent considerable time manually recording the multi-view overlaps existing start time, end time, and corresponding car IDs, and generated corresponding masks and flows (more details are shown in Appendix A). Due to time constraints, we tried our best to convert a single scene from the V2XSet dataset (ECCV 2022) into the V2XSet-N-mini dataset for our experiments. As shown in Table 1 of the attached PDF, the RCDN significantly improves the average performance of the drivable area, lane, and dynamic vehicle detection by 74.40\%, 110.69\%, and 201.74\%, respectively, compared to baseline methods without RCDN. Additionally, we conducted open-set inference on the V2X-Sim 2.0 dataset (RA-L 2022), deploying the pre-trained RCDN model from OPV2V-N directly onto the V2X-Sim 2.0 dataset. Figure 1 in the attached PDF demonstrates that RCDN effectively stabilizes the perception results.


**A1.2** **For the real-world datasets**, the existing open-source real-world collaborative perception datasets are DAIR-V2X (CVPR 2022) and V2V4Real (CVPR 2023). i) V2V4Real focuses on the two connected vehicle-to-vehicle collaboration and currently only provides LiDAR data without camera modality, targeting the downstream of 3D detection. As for our proposed RCDN, we do not introduce lidar modality just like single-agent perception to deal with the robust perception setting, we want to focus on utilizing the special property of collaborative perception: multi-view based. Hence, V2V4Real cannot be utilized for our purposes. ii) DAIR-V2X focuses on the collaboration between one road-side infrastructure (equipped with one camera and one lidar) with one connected vehicle (equipped with one camera and one lidar). During our experiments, we found that the infrastructure cameras are positioned much higher than those on the vehicle, resulting in fewer overlapping views compared to vehicle-to-vehicle collaboration. Due to these limitations, we are also unable to utilize DAIR-V2X. Theoretically, the proposed RCDN compensates for the negative impact of noisy cameras through multi-view accurate reconstruction. The distinction between synthetic and real datasets primarily affects the distribution of RGB images, influencing later BEV feature space but not conflicting with the fundamental principles of RCDN. Therefore, the effectiveness of RCDN remains unchanged whether using synthetic or real datasets. RCDN has been thoroughly validated in synthetic OPV2V-N experiments, demonstrating stable performance under noisy camera conditions. However, we will keep track of real-world dataset development, implementing the RCDN in real-world settings when the camera settings meet our requirements.

---

### Decision · Program_Chairs · 2024-09-25

**Decision:**

Accept (poster)

**Comment:**

The paper received 3 Borderline Accept and 1 Weak Accept. The reviewers agree that the solution and the problem is interesting. However, reviewers think that the evaluation is weak, but enough to show the benefits of the proposed approach. After reading the paper, the reviews, and discussion, I also think the problem and solution is interesting and can cause excitement in the community. Thus, I recommend acceptance. Nevertheless, I encourage the authors to revise the paper given the feedback the reviewers gave.